# Basic Cognition of Melatonin Regulation of Plant Growth under Salt Stress: A Meta-Analysis

**DOI:** 10.3390/antiox11081610

**Published:** 2022-08-19

**Authors:** Feiyu Yan, Hongliang Zhao, Longmei Wu, Zhiwei Huang, Yuan Niu, Bo Qi, Linqing Zhang, Song Fan, Yanfeng Ding, Ganghua Li, Guoliang Zhang

**Affiliations:** 1School of Life Sciences and Food Engineering, Huaiyin Institute of Technology, Huai’an 223003, China; 2Rice Research Institute, Guangdong Academy of Agricultural Sciences, Guangzhou 510640, China; 3College of Agronomy, Nanjing Agricultural University, Nanjing 210014, China; 4State Key Laboratory of Soil and Agricultural Sustainable Development, Nanjing 210008, China; 5Jiangsu Key Laboratory of Attapulgite Clay Resource Utilization, Huai’an 223003, China

**Keywords:** melatonin, meta-analysis, plant growth, salt stress

## Abstract

Salt stress severely restricts the growth of plants and threatens the development of agriculture throughout the world. Worldwide studies have shown that exogenous melatonin (MT) can effectively improve the growth of plants under salt stress. Through a meta-analysis of 549 observations, this study first explored the effects of salt stress characteristics and MT application characteristics on MT regulated plant growth under salt stress. The results show that MT has a wide range of regulatory effects on plant growth indicators under salt stress, of which the regulatory effect on root indexes is the strongest, and this regulatory effect is not species-specific. The intensity of salt stress did not affect the positive effect of MT on plant growth, but the application effect of MT in soil was stronger than that in rooting medium. This meta-analysis also revealed that the foliar application of a concentration between 100–200 μM is the best condition for MT to enhance plant growth under salt stress. The results can inspire scientific research and practical production, while seeking the maximum improvement in plant salt tolerance under salt stress.

## 1. Introduction

Salt stress has seriously threatened the world’s agricultural production. Poor irrigation practices and excessive chemical fertilization have led to the occurrence of secondary salinization of cultivated land, which has seriously inhibited the growth and yield of crops [1,2]. Soil salinization affects about 10.00 × 10^9^ hectares of land in the world, causing 7–8% productivity loss, becoming a key constraint to global crop production and a serious threat to food security [3]. Therefore, obtaining salt tolerance is very important for realizing the dual challenges of global food security and the sustainable development of modern agriculture [4,5]. Salt stress mainly affects plants through osmotic stress and ionic stress. Low water potential at the initial stage of plant stress leads to difficulty in water absorption by roots, resulting in water deficiency and osmotic stress similar to physiological drought. With the extension of stress duration, excessive sodium ions accumulate in cells, resulting in membrane rupture and cytoplasmic Na^+^/K^+^ ion imbalance [6]. These primary stresses will lead to secondary oxidative stresses, nutritional imbalance, and hormone imbalance. Secondary effects collectively lead to the overgeneration of reactive oxygen species (ROS) [7]. Although the production of ROS is a common phenomenon and a part of plant cell metabolism, environmental stress will lead to the excessive production of ROS. ROS are not only highly reactive, but also toxic in nature, and will damage lipids, proteins, carbohydrates and DNA [8]. Recent studies have shown that ROS plays a dual role in plants; whether they act as signaling molecules or as stressors depends on the critical balance between ROS production and clearance [9]. However, excessive ROS accumulation is always harmful to plants; through the combined action of primary stress and secondary stress, salt stress affects the physiological activities of plants, such as photosynthesis, carbon and nitrogen metabolism, respiration, and protein synthesis. The changes of plant physiological activities under salt stress lead to the slowdown of plant growth. Under the condition of salt stress, the ion-independent growth decreased in a few minutes to a few days, which mainly led to the inhibition of cell expansion in shoots. The accumulation of salt stress injury on plant physiological activities will further lead to the decline in growth parameters [10,11,12]. Salt treatment resulted in a significant decrease in plant height, root length and dry weight, the fresh and dry weight of cotton seedlings [13], and the seed germination rate of wheat varieties [14]. After salt stress, the growth inhibition of resistant varieties was significantly lower than that of sensitive varieties [15,16]. Taking rice as an example, under 100 mM NaCl stress, the dry weight of resistant varieties decreased by 19.23%, while that of sensitive varieties decreased by 49.38% [17]. Tomato and pepper plants grown under salt stress experienced a decline in dry weight and plant height [18,19]. In addition, germination of seeds faces serious limitations upon exposure to salt stress. Root growth is more susceptible to salt conditions than stem growth, but both are affected, which is a reasonable index of salt damage. Root and shoot lengths are two indicators of rice plant response to salt stress [20]. The change in plant growth parameters is the most intuitive expression of plant stress growth.

Plant hormones are signaling molecules that play a crucial role in plant growth and development [21]. Applying exogenous plant hormones to salt stressed plants can alleviate the negative effects of salt stress on physiological activities and growth. The use of plant hormones to improve plant salt tolerance has also received extensive attention [22]. Melatonin (MT) is a small molecule substance with multiple regulatory effects in plants and animals. It was first discovered in the hypothalamus of cattle in 1958 and also in plants in the 1990s [23]. Recently, the identification of the first putative plant MT receptor in *Arabidopsis* suggests that MT could be a plant hormone [24,25]. MT enhances plant tolerance to salt stress in two ways. One is through direct impact—the molecular structure of MT enables it to provide electrons to ROS. Unlike other traditional antioxidants, MT molecules can scavenge up to eight or more ROS through the free radical scavenging cascade [26]. MT can also indirectly scavenge ROS by activating antioxidant enzymes and the glutathione-ascorbate cycle, and increasing the expression of antioxidant enzyme genes under stress conditions [27]. The other is through indirect impact—by enhancing the activity of antioxidant enzymes, photosynthetic efficiency, and metabolite content, and by regulating stress-related transcription factors [28]. In addition, MT can affect plant performance by affecting gene expression. Other precursors and metabolites related to MT, such as tryptophan and 5-hydroxylas, can also improve the tolerance of plants to salt stress [29]. Salt stress results in the rapid accumulation of MT in plants [30]. Recent studies have shown that MT plays an important role in regulating plant reactive oxygen species and reactive nitrogen signals under salt stress [31,32].

In view of its excellent antioxidant capacity and the ability to regulate plant growth, exogenous MT has been widely reported to enable plants to cope with a variety of stresses [33,34]. However, there are still many basic points that have not been clarified, which leads to the fact that MT regulates plant salt tolerance in the laboratory, but it is difficult to evaluate the actual production. Studies have shown that MT has the ability to effectively alleviate salt stress in food crops [35], horticultural crops [36], and cash crops [37], but no study has reported whether there is species-specific regulation of plant salt tolerance by MT. Similarly, MT has been reported to widely regulate plant growth parameters under salt stress [38], including dry weight, fresh weight, plant height, root length, leaf area, and root surface area. However, it is not clear whether MT has a specific regulatory effect on growth parameters. MT plays an important role in the regulation of plant growth and development. At the same time, MT is also considered an important regulator for plants to cope with stress, and some studies have discussed the function of MT as a plant growth regulator or a plant stress stimulant [39], but did not clearly explain which of the above two functions MT focuses on. MT has been used to enable plants to cope with various levels of salt stress [40], but there is no report exploring whether the regulation effect of MT is affected by stress intensity. MT has been applied to plants through seed soaking, spraying, seed priming, and other methods [40,41], but there is no study on the effects of various application methods. Similarly, the optimal concentration of MT for regulating plant growth under salt stress has not been reported. Even for the same plant, the concentration difference of MT applied by different scholars can reach 200 times [42,43]. These problems make it difficult for MT to move from the laboratory to actual production.

Meta analysis is a quantitative statistical model which systematically analyzes the complex factors that affect related dependent variables [44]. Since these variables are conclusions drawn from published articles, meta-analysis improves the objectivity of the systematic evaluation of independent empirical research. Therefore, this study used meta-analysis methods to explore whether there is species specificity in MT regulating plant growth under salt stress, to explore the effects of salt stress characteristics on MT regulation of plant growth under salt stress, and to determine the best application method and concentration of MT to improve plant growth under salt stress. The results of this meta-analysis are of great significance to guide the regulation of plant salt tolerance by MT. At the same time, the results presented in this study highlight the areas that need further research.

## 2. Materials and Methods

### 2.1. Literature Survey and Selection

This is a meta-analysis study on melatonin regulating plant growth under salt stress. In this study, we followed the PRISMA 2020 statement [45]. On 15 March 2022, different scientific literature tools (Web of Science, Science Direct, Springer Link, and Pubmed) were used to search for various keywords (melatonin, phytomelatonin, salinity, salt stress, salt tolerance, plants), focusing on the regulation of exogenous MT on plant growth parameters under salt stress. The specific retrieval process is shown in Figure 1. The search yielded 1515 English articles. Articles were included or excluded according to a three-step procedure. In the first step, we first excluded duplicate articles through the literature management software (Endenote X9, Thomson ResearchSoft, Stanford, CA, USA), and then manually excluded duplicate articles, leaving 302 articles. After that, we deleted the non-original research papers, and determine whether the main body of the article included literature related to the MT regulation of plant growth under salt stress according to the title, abstract and key words, and screening parameters, retaining 115 papers. In the second step, the remaining articles were screened based on whether they met the following conditions: the degree of salt stress remained stable during the experiment, exogenous MT treatment was used, repetition was set in the experiment, data dispersion was reported, and the article provided any of the plant growth parameters after MT treatment, leaving 42 qualified articles (Table A1). In the third step, data related to growth parameters was extracted and analyzed. The literature survey and selection were conducted manually by two reviewers, working independently under the supervision and guidance of the lead author; no workflow management platform/software was used.

### 2.2. Data Extraction

In each article, the average value, error term (SD or SE), sample size (n), and replication of growth parameters of plants in the control group and exogenous MT treatment group were recorded in the database. For the data in text and tables, we directly extracted these. For the data in graphs, we used data extraction software (WebPlotDigitizer, version 4.5, https://automeris.io/WebPlotDigitizer/, accessed on 1 May 2022) [46] to extract the digitized value, with an accuracy of ±1% relative to the actual value. For the measured values at different times of the same test, we only included the results of the last measurement. For the study of multiple MT concentration treatments and a common control, or different levels of salt stress set in the same study, we regarded these as statistically independent, and we extracted the data separately [47].

In addition, we also extracted the plant species in each study, the concentration of sodium chloride used to produce salt stress, the method of salt stress application, the concentration of exogenous MT, the method of exogenous MT application, and the type of growth parameter indicators used to evaluate plant salt tolerance. Based on the above extraction rules, we extracted 549 observations from 42 articles.

### 2.3. Meta-Analysis

#### 2.3.1. Effect Sizes

The meta-analysis was performed using STATA 17 software (http://www.stata.com, accessed on 1 May 2022). In order to study the regulatory effect of exogenous MT on plant growth parameters under salt stress, a random effect meta-analysis was used. We used the natural log of the response ratio (LnR) as a measure of the effect size (ES) [48]:lnR = ln(X_t_/X_c_) = lnX_t_ − lnX_c_
where X_t_ and X_c_ are the measured values of plant growth parameters under MT treatment and MT-free treatment, respectively.

The variance (V) of the corresponding index effect value (lnR) was calculated as [49]:V = S_t_^2^/N_t_X_t_^2^ + S_t_^2^/N_t_X_t_^2^
where St is the standard deviation of plant growth parameters with MT, Sc is the standard deviation of plant growth parameters without MT, Nt is the sample size of plant growth parameters with MT, and Nc is the sample size of plant growth parameters without MT.

Subsequently, the log response ratios were combined across the studies using a weighting procedure. The mean effect size (ln$\bar{R}$) was calculated as follows [50]:$$\ln\bar{R}=\sum_{i=1}^{k}W_{i}\ln R_{i}/\sum_{i=1}^{k}W_{i}\;\;\;\;\;W_{i}=1/V_{i}$$
where lnR_i_ is the average effect size; W_i_ is the weight of i, that is the reciprocal of sample variance vi; LnR_i_ is the logarithmic response ratio of i; and k is the number of statistical studies.

#### 2.3.2. Heterogeneity Test

During meta-analysis, it is necessary to carry out a heterogeneity test (q test) on the collected relevant data to test whether there is heterogeneity in the sample data. If *p* > 0.05 (chi-squared distribution test), it indicates that there is no heterogeneity in the data, and the fixed effect model is selected; if *p* < 0.05 (chi-squared distribution test), it indicates that the data is heterogeneous; therefore, the random effect model should be used and subgroup analysis should be conducted to explore the source of heterogeneity [51].

In this study, the bias test was conducted on the collected yield data. The frequency distribution of the effect value of plant growth parameters was fitted to a Gaussian function. According to the K–S (Kolmogorov–Smirnov) test (Figure A1), the frequency distribution of the effect value of plant growth parameters did not obey the normal distribution (*p* < 0.01). Therefore, this paper uses the comprehensive effect value (ln$\bar{R}$) and 95% confidence interval generated by the nonparametric estimation method (bootstrapping) for data analysis.

#### 2.3.3. Meta Subgroup Analysis

For better identifying the potential interactive effects, plant species were divided into 18 categories, and salt stress conditions were divided into salt stress and normal condition. We also grouped them according to the concentration of NaCl that produces salt stress. The application of salt stress was grouped into two types, exogenous MT concentrations were divided into nine groups, MT was treated in four ways, and the growth parameters were divided into four categories. See Table 1 for specific categories.

Through meta subgroup analysis, the explanatory variables of each group were studied. The results showed that, except for the presence or absence of salt and the type of growth parameters, other grouping methods significantly (*p*-value < 0.0001) affected the MT regulation of plant growth under salt stress (Table 2).

#### 2.3.4. Publication Bias Test

To clarify the publication bias, the fail-safe numbers were presented. A fail-safe number N > 5n + 10 (N = 1,775,954.1, n = 549) indicated that the result had no publication bias [52]. At the same time, we also conducted Egger’s test (Table A2), and the result showed that *p* < 0.05, which indicates that there may be publication bias. Therefore, we carried out the nonparametric trim-and-fill analysis of publication bias, and the results showed that adding another 104 studies had no significant impact on the results (Figure A2, Table A3), indicating that the results of this meta-analysis were effective.

### 2.4. Statistics

The meta-analysis was performed using STATA 17 software (StataCorp LLC, College Station, Texas, USA). Graphics were drawn using STATA or Graphpad prism (Graphpad Software Inc., San Diego, CA, USA).

## 3. Results

### 3.1. Overall Response of Plant Growth to Exogenous MT

According to the collected studies, compared with no application of exogenous MT, the application of MT significantly improved the plant growth parameters, and the comprehensive effect size was 0.178 (95% CI: 0.154–0.202). According to the contour map, a low concentration of exogenous MT (0–50 μM) had a weak regulatory effect on growth parameters. The surface fitted with a quintic equation of all data also showed similar results: a high concentration of MT had a strong positive regulatory effect on growth parameters (Figure 2).

### 3.2. The Difference in Response of Growth Parameters to Exogenous MT

Exogenous MT significantly (*p* < 0.05) promoted other plant growth parameters, except for stem diameter, root diameter, and root surface. The effect value of the root dry weight group was the largest (0.305, CI: 0.228–0.382) (Figure 3A). MT had the strongest effect on root indexes (ES: 0.219, CI: 0.181–0.258) and the weakest effect on whole plant indexes (ES: 0.076, CI: −0.033–0.185) (Figure 3B).

### 3.3. The Response of Plant Species to Exogenous MT

Among all the plants included in the study, MT significantly (*p* < 0.05) promoted the growth parameters of 12 plants (*Avena nuda* (naked oat), *Cucumis melon* (melon), *Cucumis sativus* (cucumber), *Dracocephalum kotschyi* (Savannah obedient-plant), *Gossypium hirsutum* (cotton), *Limonium bicolor* (statice bicolor), *Medicago sativa* (alfalfa), *Oryza sativa* (rice), *Phaseolus vulgaris* (common bean), *Solanum lycopersicum* (tomato), *Triticum aestivum* (wheat), *Zea mays* (maize)), but not *Avena sativa* (oat), *Brassica napus* (rapeseed), *Helianthus annuus* (sunflower), *Hordeum vulgare* (barley), *Malus hupehensis* (tea crabapple) and *Stevia rebaudiana Bertoni* (candyleaf). Among these, MT had the strongest regulatory effect on the growth of Avena nuda (ES: 0.322, CI: 0.181–0.463) and the weakest regulatory effect on *Gossypium hirsutum* (ES: 0.114, CI: 0.06–0.0.167). The fluctuation range of effect value was small, and MT did not show specificity in regulating the growth of the plants included in the study (Figure 4).

### 3.4. The Effect of Salt Stress Intensity and Rooting Environment

The regulatory effect of MT on the growth of plants growing in soil is significantly stronger than that of plants growing in rooting medium (Figure 5A). Whether under salt stress or a normal environment, exogenous MT showed a significant promoting effect on plant growth, and there was no significant difference between the two environments. After grouping for salt stress, it was found that when the concentration of sodium chloride was 250 mM–300 mM, the promoting effect of MT on plant growth was significantly higher than that at other concentrations (Figure 5C). There was no significant difference between the effect values of other concentrations.

### 3.5. The Effect of MT Concentration and Application Method

When the concentration of MT was greater than 1 μM, MT showed a significant effect on the growth of plants under salt stress, and this effect increased with the increase in concentration. The effect value of the 300–400 μM group was the largest, which was 0.763 (CI: 0.371–1.154), but when the concentration reached 400–500 μM, the effect value showed a downward trend (Figure 6A). Under various application methods, MT showed significant effects on the regulation of plant growth under salt stress. The effect value of MT added to the culture medium by seed soaking and spraying was significantly higher than that of the seed priming group, and the effect value of the spraying group was the largest (0.252, CI: 0.217–0.288) (Figure 6B).

## 4. Discussion

### 4.1. Effect of MT on Plant Growth under Salt Stress

MT has a wide range of regulatory effects on the growth of different plants (Figure 4), and the content of endogenous MT in many plants is significantly increased after being treated with exogenous MT, which indicates that there is a common response mechanism of exogenous MT in plants [53,54]. Salt stress also increased the content of endogenous MT in plants. MT showed a significant positive regulatory effect on the growth parameters and germination indexes of plants under salt stress (Figure 2 and Figure 3).

However, the subgroup analysis showed that MT had the strongest regulatory effect on root indexes (Figure 3B). The root is the first plant organ that contacts, senses, and responds to soil salinity [55], playing an important role in plants’ response to salt stress. Compared with aboveground parts, the root system is less inhibited by salt [56]. Although moderate salt stress may stimulate the growth of roots, higher stress levels are usually related to the decline in root development. The developed root system is conducive to the absorption of water and nutrients, and sodium ions are mainly discharged through the root tip SOS pathway [57]. The developed root system is also conducive to the discharge of toxic sodium ions [58]. The results showed that salt tolerant olive varieties could maintain higher fine root biomass and increased water conductivity under salt stress [59]. Similarly, salt tolerant rice, wheat, and maize also have larger root biomass and surface area [60,61,62]. Extensive studies have shown that MT is a key molecule for root growth and development [63]; this is mainly because MT can reduce ionic toxicity, osmotic stress, and oxidative stress, thereby enhancing the antioxidant system, photosynthesis, and glucose metabolism [64].

Our previous research results showed that MT not only increased the biomass of rice roots, but also promoted the Na^+^ efflux and potassium ion internal flow in roots [65], and alleviated the oxidative damage of roots [66]. The results showed that the most abundant MT response genes in the root system of *Phaseolus vulgaris* under salt stress were mainly involved in transcription regulation, the redox process, transcription factor activity, and oxidoreductase activity [67]. MT regulates caspase-3-like activity and programmed cell death-related gene expression in alfalfa roots under salt stress [68]. Hydrogen sulfide can alleviate the inhibition of root growth caused by salt stress, which can be enhanced by MT because MT regulates hydrogen sulfide homeostasis and L-cysteine desulfurase activity [69]. The results of the subgroup analysis also showed that exogenous MT had a significant positive regulatory effect on germination indexes (Figure 3B). Seed germination is a key stage in the plant life cycle. At this stage, static seeds are transformed into highly active seedlings. Most plant species are sensitive to salt stress at the early stage of seed germination and seedling establishment [70]. The results showed that MT regulated the expression of abscisic acid (ABA) and gibberellin (GA) genes in the plant signal transduction pathways, induced cotton seed radicle development and germination, and alleviated dormancy [13]. Similar results were obtained in the study of *Limonium bicolor*. The seeds pretreated with MT contained high levels of MT and GA, low levels of ABA, and high levels of amylase and α-amylase activity. MT also plays a regulatory role in GA and ABA biosynthesis-related genes [41]. The results showed that the activities of antioxidant enzymes, including superoxide dismutase (SOD), catalase (CAT), and peroxidase (POD), in MT pretreated seeds significantly increased by about 1.3–5.0 fold, while the *Cu-Zn SOD*, *Fe-Zn SOD*, *CAT*, and *POD* increased by 1.4–2.0 fold [71]. In general, MT showed a wide range of regulatory effects on plant growth indicators under salt stress, but the results of this meta-analysis showed that MT had the strongest regulatory effect on root-related indicators.

### 4.2. Effects of Salt Stress Characteristics on MT Regulation of Plant Growth under Salt Stress

Various characteristics of salt stress significantly affect the salt tolerance of plants. For example, the higher the salt concentration, and the longer the exposure to salt stress, the worse the tolerance of plants, and the more serious the damage [72]. Similarly, salt characteristics can also affect the regulatory effects of exogenous hormones or regulators on plant stress tolerance. The results showed that under 50 mM salt stress, the shoot dry weight of pistachio plants treated with MT increased by 16.6% compared with that without MT treatment, but when the salt stress concentration increased to 150 mM, the increase rate was 32.8% [73]. In many plants, it has been observed that the absolute effect of MT on regulating growth changes with the increase in salt stress [40,74], but this phenomenon has not been systematically studied. The results of the subgroup analysis in this study show that the application mode of salt stress (soil or rooting medium) affects the effect of MT on regulating plant growth under salt stress. When MT is applied to plants planted in soil, its regulation effect is significantly greater than that applied to plants in rooting medium (Figure 5A). These observations may indicate that MT applied to field crops is more effective than MT applies to hydroponic crops in the factory, but so far, there is no relevant report regarding this issue. We infer that this may be due to more environmental stress factors in the field and the greater potential for crop growth improvement [75]. Factory hydroponics can precisely control the growth conditions of plants [76], and there is little room for plant growth to be improved. This hypothesis requiures further research to explore and confirm.

In the initial research, MT, an endogenous growth regulatory signal in crops, has attracted much attention [29]. With the deepening of research, some scholars have discussed the function of MT as a plant growth regulator and biological stimulant [24,39,77]. They have focused on the MT regulation of the growth of roots, shoots, and explants to activate seed germination and rooting. The antioxidation of MT partially explains the fact that it enhances the ability of plants to withstand abiotic stress [78], and scholars believe that MT is a candidate drug for the natural biological stimulation treatment of field crops [79]. In recent years, the discussion about MT has mainly focused on the fact that MT provides a wide range of stress regulation in plants and the perfection of the evidence of MT as a plant hormone [27,80,81]. Based on the results of this study, we believe that MT does not show stress specificity in the function of regulating plant growth; that is, the relative effect of MT on plant growth regulation is not affected by the presence of intensity of salt stress (Figure 5). This means that MT can be used to promote plant growth under normal growth conditions and improve plant tolerance under adverse conditions.

Extensive studies have also proved this view. Melatonin plays an active role in regulating plants to cope with oxidative stress [26], drought stress [82], and chilling stress [83]. However, the existing studies pay less attention to the practical application factors of MT, such as concentration and application methods, and have not formed a consistent opinion.

### 4.3. Suitable Concentration Range and Application Method of MT in Regulating Plant Growth under Salt Stress

The use of MT to regulate plant growth under salt stress has been widely studied. Different application methods and concentrations have been reported to effectively improve plant growth under salt stress, but the application characteristics of MT in different reports are quite different. For example, for the seedling stage of cotton, some studies reported that 0.1 μM [30] showed a significant growth promoting effect. However, some studies have shown that it still has the effect of promoting growth when the concentration is as high as 200 μM [84]. For different crops, the reported effective concentration difference is as high as hundreds of times that level. This large span of effective concentration indicates the universality of MT in regulating plant growth, but it also generates problems regarding the application of MT in actual production. The results of the subgroup analysis in this study showed that the effect value of MT regulation of plant growth under salt stress increased with the increase in MT concentration, and the maximum effect value appeared in the MT concentration of 300–400 μM, but when the MT concentration was greater than 100 μM, there was no significant difference between the effect values of each concentration (Figure 6A). In addition, high concentrations of plant hormones will inhibit growth. Considering the cost of MT application in actual production, we believe that the appropriate concentration of MT for regulating plant growth under salt stress is 100–200 μM; excessive concentration may lead to the occurrence of an inhibitory effect and also incur huge application costs.

The subgroup analysis showed that the effect value of the seed priming treatment group was the smallest (Figure 6B). However, seed priming is still a seed treatment technology worthy of promotion because seed priming with plant hormones can improve plant growth and yield and reduce the impact of environmental pressure [85]. However, this innovation has received little attention from farmers. MT priming has also been used to improve the salt tolerance of cotton [86], drought tolerance of rapeseed [82] and cold tolerance of maize [83]. However, MT concentration and start-up duration may vary from crop to crop. Excessive initiation of seeds for a longer time may lead to drying and decomposition of seeds, or inactivation of seeds due to bacterial infection [87]. Future studies need to further clarify the pathways and effects involved in priming duration and MT concentration.

Studies have shown that when MT is applied to improve the cold tolerance of rice, the effect of spraying is weaker than that provided by soaking seeds and immersing roots [88]. On the other hand, the subgroup analysis results of this study showed that the effect value of the spraying treatment group was the highest (Figure 6B), which indicated that spraying might be the best way to apply MT in actual production. However, future research needs to further study the absorption and transport mechanism of plant exogenous MT, which would be conducive to providing theoretical guidance for bringing MT from the laboratory into production. The research in this area is still blank at present. In general, the results of this study show that spraying MT at a concentration of 100–200 μM is the most effective way to improve plant growth under salt stress.

## 5. Conclusions

Based on 549 observations from 42 articles, this meta-analysis has shown that exogenous MT can effectively improve plant growth under salt stress. MT has a wide range of regulatory effects on plant growth indicators under salt stress, of which the regulatory effect on root indicators is the strongest, and this regulatory effect is not species-specific. The intensity of salt stress did not affect the positive effect of MT on plant growth, but the application effect of MT in soil was stronger than that in rooting medium. This meta-analysis also revealed that the foliar spraying application of a concentration between 100–200 μM is the best condition for MT to enhance plant growth under salt stress. These results are of great significance to promote the application of MT in practical production. The research results can inspire scientific research and practical production, while seeking the maximum improvement of plant salt tolerance under salt stress. However, most of the current studies focus on the direct regulation of plant growth by MT, and there are few studies on indirect regulation pathways, such as soil microorganisms. In addition, there are no reports on the economic benefits of the MT regulation of plant salt tolerance. The solution to these problems in the future would be conducive to regulating plant salt tolerance through moving MT treatment from the laboratory to production.

## Figures and Tables

**Figure 1 antioxidants-11-01610-f001:**
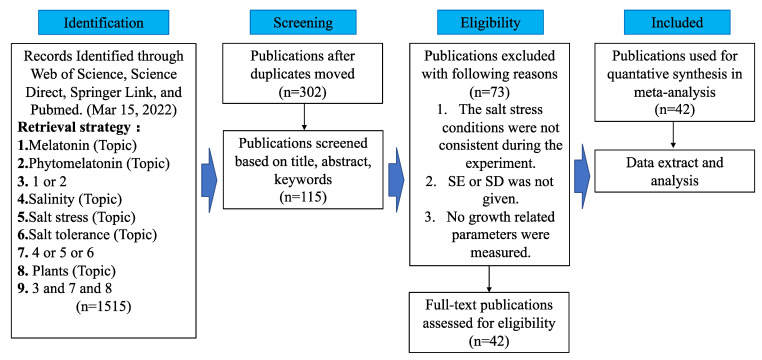
Flow chart for selection of publications.

**Figure 2 antioxidants-11-01610-f002:**
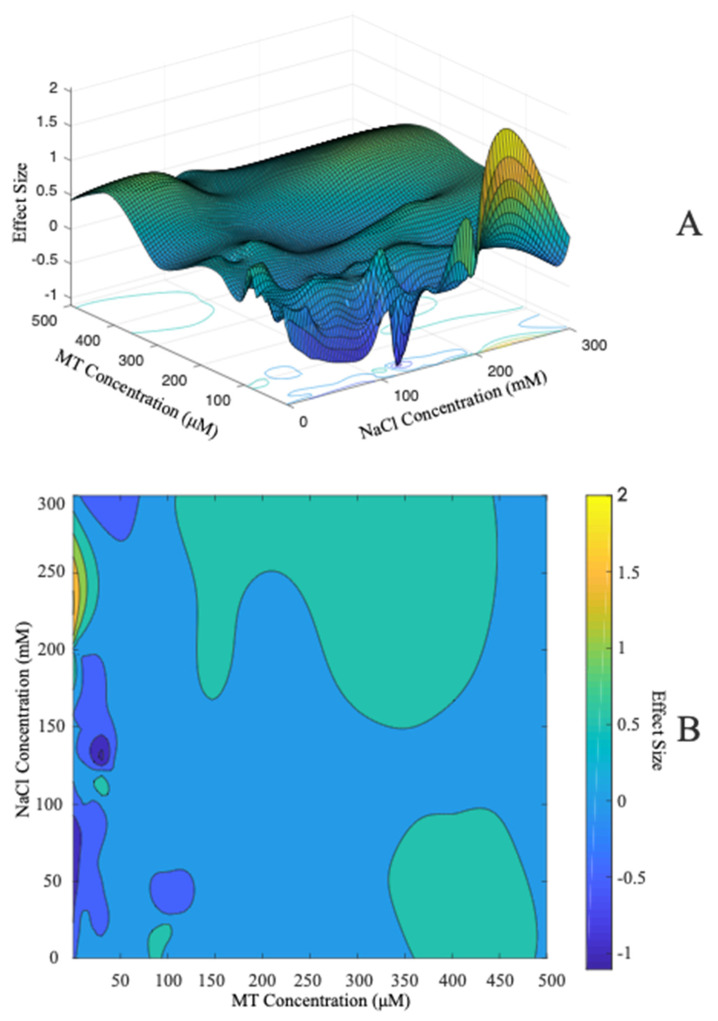
Effects of MT and salt stress on plant growth. (**A**) shows the surface without fitting, and (**B**) shows the contour map of (**A**). We also carried out quintic equation fitting on the data, and the fitted equation and surface diagram are shown in Figure A3.

**Figure 3 antioxidants-11-01610-f003:**
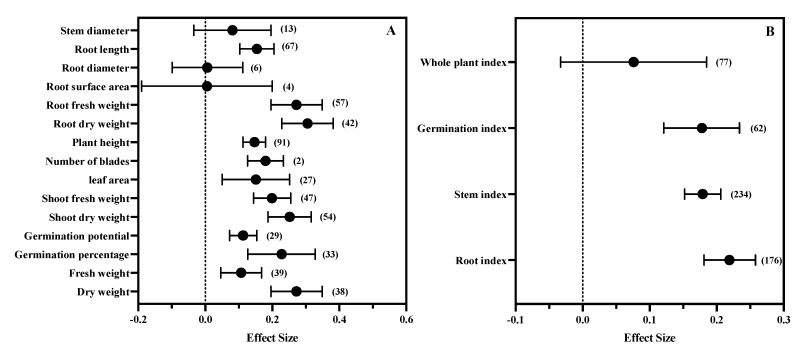
The difference in response of growth parameters. (**A**) Growth parameters; (**B**) classification of growth parameters. The symbol represents the effect size, and the bars show the 95% bootstrapped confidence intervals. The values in the parentheses show the number of observations.

**Figure 4 antioxidants-11-01610-f004:**
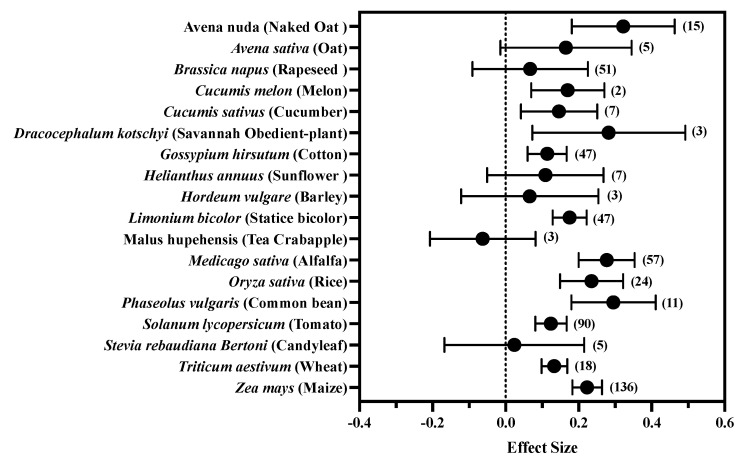
Effect of exogenous MT on plant growth under salt stress. The symbol represents the effect size, and the bars show the 95% bootstrapped confidence intervals. The values in the parentheses show the number of observations.

**Figure 5 antioxidants-11-01610-f005:**
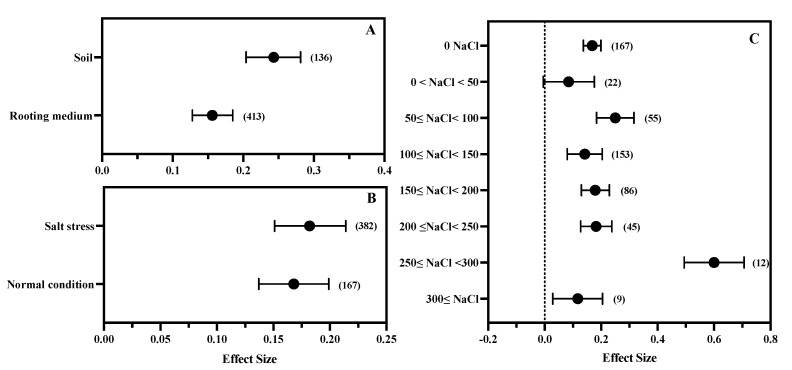
The effects of rooting environment (**A**), salt stress (**B**), and salt stress concentration (**C**) on MT regulation of plant growth under salt stress. The symbol represents the effect size, and the bars show the 95% bootstrapped confidence intervals. The values in the parentheses show the number of observations.

**Figure 6 antioxidants-11-01610-f006:**
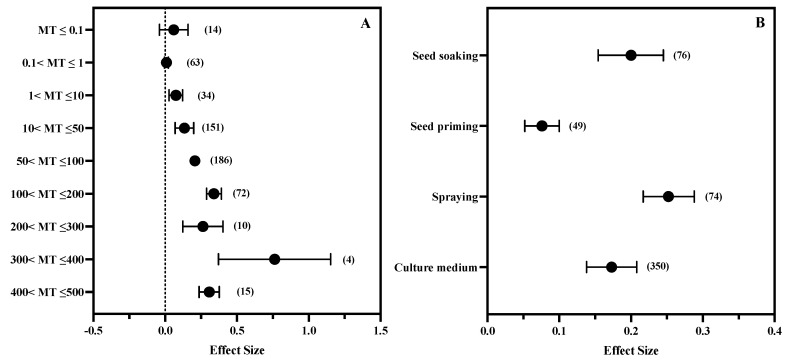
Effects of MT concentration (**A**) and application method (**B**) on the protective effects of MT in plants. The symbol represents the effect size, and the bars show the 95% bootstrapped confidence intervals. The values in the parentheses show the number of observations.

**Table 1 antioxidants-11-01610-t001:** Factors categorized as predictive variables in this meta-analysis.

Influence Factors		Classification of Subgroup
Plant properties	Plant species	Avena nuda (naked oat), *Avena sativa* (oat), *Brassica napus* (rapeseed), *Cucumis melon* (melon), *Cucumis sativus* (cucumber), *Dracocephalum kotschyi* (Savannah obedient-plant), *Gossypium hirsutum* (cotton), *Helianthus annuus* (sunflower), *Hordeum vulgare* (barley), *Limonium bicolor* (statice bicolor), *Malus hupehensis* (tea crabapple), *Medicago sativa* (alfalfa), *Oryza sativa* (rice), *Phaseolus vulgaris* (common bean), *Solanum lycopersicum* (tomato), *Stevia rebaudiana Bertoni* (candyleaf), *Triticum aestivum* (wheat), *Zea mays* (maize)
	Growth parameters	Stem diameter, Root length, Root diameter, Root surface area, Root fresh weight, Root dry weight, Plant height, Number of blades, Leaf area, Shoot fresh weight, Shoot dry weight, Germination potential, Germination percentage, Fresh weight, Dry weight.
	Types of growth parameters	Whole plant index, Germination index, Stem index, Root index.
Salt stress properties	Salt stress or not	Salt stress, Normal condition
	Salt stress (mM NaCl) concentration	0 = NaCl, 0 < NaCl < 50, 50 ≤ NaCl < 100, 100 ≤ NaCl < 150, 150 ≤ NaCl < 200, 200 ≤ NaCl < 250, 250 ≤ NaCl < 300, 300 ≤ NaCl.
	Ways of applying salt stress (rooting environment)	Rooting medium, Soil.
Meltonin properties	MT concentration (μM)	MT ≤ 0.1, 0.1 < MT ≤ 1, 1 < MT ≤ 10, 0 < MT ≤ 50, 50 < MT ≤ 100, 100 < MT ≤ 200, 200 < MT ≤ 300, 300 < MT ≤ 400, 400 < MT ≤ 500.
	Mode of exogenous application	Rooting medium, Foliar application, Seed priming, Seed soaking

**Table 2 antioxidants-11-01610-t002:** Heterogeneity and *p*-value for the direct seeding effect size on growth across different categorical variables.

Categorical Variable	QB	df	*p*-Value
Plants	54.83	17	0.000
Salt stress or not	0.40	1	0.526
Type of rooting environment	12.48	1	0.000
Salt concentration	72.62	7	0.000
MT concentration	335.08	8	0.000
MT treatment type	74.42	3	0.000
Growth parameters	56.00	14	0.000
Types of growth parameters	7.15	3	0.067

QB = the statistic value of heterogeneity for explaining variance; df = the degree of freedom; *p*-value = the significant value of heterogeneity for explaining variance.

## Data Availability

Data contained within the article.

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
