# Peer review of "Basic Cognition of Melatonin Regulation of Plant Growth under Salt Stress: A Meta-Analysis"

_antioxidants, 2022, doi:10.3390/antiox11081610_

Round 1

Reviewer 1 Report

The manuscript concerns an interesting and up-to-date topic of the possible application of melatonin in alleviation of salt stress toxicity. The findings are based on meta-analysis of published articles. In my opinion, the meta-analysis have great value as the amount of information increased gradually. The methods of data search, screening and final selection is well presented and justified. The articles selected for the meta-analysis are published in reputative, international, peer-reviewed journals. The manuscript is written in a clear and easy to follow manner and the conclusions are justified by the results.

Some minor comments/suggestions/questions:

11)    Line 21: should be “species-specific” instead of “specific-specific”?

22)  Lines 81-83: I would suggest to include the term “indirectly” already in the sentence “MT can also scavenge ROS…”. Otherwise there is a logical inconsistency between the two sentences.

33)  Line 105: I would suggest to change the term “MT prefers”

44)  Lines 91-113: It would be good to include all the questions on MT effects in plant in additional short table.

55)  In Fig. 1 first box – point nr 3 – I would suggest to include wider spaces between “1”  and “or”

66)  Line 160: maybe better “the method of salt stress application”?

77)  Table 1: The species names in Latin should be written in  italic. It would be good to include also English names of the plants.

88)  Fig. 2: The “A” and “B” description is missing. I would suggest to increase slightly the size of Fig. 2A.

99)  Lines 256- 261: In my opinion it would be good to sill list the plants affected by the MT and list of plants not affected in the text, despite the fact that it is seen in the graph. I would also suggest to include the English names, as sometimes the species are better recognized that way.

110) Line 292: I would suggest to change the title of the subsection 4.1 as the content is more focused on the impacts on growth that on species specificity.

111) Line 331: The species name should be written in italic.

112) Line 346: “The results showed that under 50 mM salt stress, (…) with MT increased by 16.6% compared with the control”. Was it control or salt stressed but MT-untreated plants?

113) Line 354-355: Could you try to explain why the effect is stronger in soil-grown than in hydroponically-grown plants?

114) Line 368-369: Could you add some examples of MT promoting effects in response to other stress factors?

115) I would suggest to include in the discussion 1-2 sentences on the economical aspect of the application of MT as a biostimulant.

Author Response

Reviewer 1

The manuscript concerns an interesting and up-to-date topic of the possible application of melatonin in alleviation of salt stress toxicity. The findings are based on meta-analysis of published articles. In my opinion, the meta-analysis have great value as the amount of information increased gradually. The methods of data search, screening and final selection is well presented and justified. The articles selected for the meta-analysis are published in reputative, international, peer-reviewed journals. The manuscript is written in a clear and easy to follow manner and the conclusions are justified by the results. 

Response: Thank you for your work in the process of reviewing the manuscript. Your comment is highly appreciated. We did our best to the revised manuscript.

Some minor comments/suggestions/questions:

1) Line 21: should be “species-specific” instead of “specific-specific”? 

Response: Thank you for your suggestion. We apologize for the mistake caused by our negligence. In the revised manuscript, we corrected this error.

2) Lines 81-83: I would suggest to include the term “indirectly” already in the sentence “MT can also scavenge ROS…”. Otherwise there is a logical inconsistency between the two sentences.

Response: Thank you for your suggestion. In the revised manuscript, we modified it according to your suggestion.

3) Line 105: I would suggest to change the term “MT prefers” 

Response: Thank you for your suggestion. In the revised manuscript, we modified it according to your suggestion.

4) Lines 91-113: It would be good to include all the questions on MT effects in plant in additional short table.

Response: Appreciated your brilliant comments. In recent years, the regulation of plant growth, development and stress tolerance by melatonin has received extensive attention. The problems related to the role of melatonin in plants have been constantly discovered and solved. There have been many excellent reviews to discuss these problems. In this study, we hope to use the method of meta-analysis to clarify some basic cognition of melatonin regulating plant growth under salt stress. Therefore, our focus is more on the regulation of melatonin on plant stress, which is more targeted. We hope that the revised manuscript will not add more information about the unresolved problems of melatonin in plants, and we hope to get your permission. In the revised manuscript, we enriched the relevant introduction. Thank you again for your work.

5) In Fig. 1 first box – point nr 3 – I would suggest to include wider spaces between “1”  and “or”

6) Line 160: maybe better “the method of salt stress application”?

7) Table 1: The species names in Latin should be written in  italic. It would be good to include also English names of the plants. 

8) Fig. 2: The “A” and “B” description is missing. I would suggest to increase slightly the size of Fig. 2A. 

Response: To make it simple and clear, we reply to the suggestions 5-8 altogether. Thank you for your suggestions. We accepted your useful suggestions. We corrected these mistakes in the resubmitted manuscript.

9) Lines 256- 261: In my opinion it would be good to sill list the plants affected by the MT and list of plants not affected in the text, despite the fact that it is seen in the graph. I would also suggest to include the English names, as sometimes the species are better recognized that way. 

Response: Thank you for pointing this out. In the revised manuscript, we list the plants affected by melatonin. We also added the English names of plants figure 4.

10) Line 292: I would suggest to change the title of the subsection 4.1 as the content is more focused on the impacts on growth that on species specificity. 

Response: Thank you for your suggestion. In the revised manuscript, we modified it according to your suggestion.

11) Line 331: The species name should be written in italic. 

Response: We are sorry for these mistakes due to our negligence. We have corrected this error in the revised manuscript.

12) Line 346: “The results showed that under 50 mM salt stress, (…) with MT increased by 16.6% compared with the control”. Was it control or salt stressed but MT-untreated plants?

Response: We apologize for our inappropriate statement. The control here refers to plants under the same salt stress condition but not treated with melatonin. We have corrected this error in the revised manuscript.

13) Line 354-355: Could you try to explain why the effect is stronger in soil-grown than in hydroponically-grown plants?

Response: Appreciated your brilliant comments. We have not found relevant reports in the existing studies. We have made a scientific hypothesis that under the field conditions, the growth of plants is faced with many adverse environmental factors, and plants have great growth potential. However, under the hydroponic or facility cultivation conditions, all the conditions required for plant growth are controlled in the optimal range, and the growth potential of plants can be brought into full play. Therefore, the effect of promoting crop growth through hormone regulation is limited. We added discussion on relevant contents in the revised manuscript.

14) Line 368-369: Could you add some examples of MT promoting effects in response to other stress factors?

Response: Thank you for your suggestion. In the revised manuscript, we added relevant content.

15) I would suggest to include in the discussion 1-2 sentences on the economical aspect of the application of MT as a biostimulant.

Response: Thank you very much for your suggestion, which is very farsighted. If melatonin is used to regulate plant growth from laboratory to actual production, the economic benefits must be paid attention to, but there is no relevant report. Our previous study showed that when melatonin was sprayed at the US $27 per hectare of paddy field, the final increase in income can reach $61.5. Of course, this result is based on the premise that melatonin reagent is not commercially produced. If commercial production of melatonin can be realized in the future, the economic income will be further improved. In the revised manuscript, we listed issues related to economic benefits as scientific issues to be studied and solved in the future.

Reviewer 2 Report

Dear authors,

Manuscript antioxidants-1855178 entiteled "Basic cognition of melatonin regulation of plant growth under salt stress: A meta-analysis" and authored by Feiyu Yan , Hongliang Zhao , Longmei Wu , Zhiwei Huang , Yuan Niu , Bo Qi , Linqing Zhang , Song Fan , Yanfeng Ding , Ganghua Li * , Guoliang Zhang targets a hot topic that is potentially very interesting for the journal readers, the whole scientific community and end users. While the work seems to be accurately conducted several issues needs attention of the authors that needs to provide clear answers that will determine the fate of the manuscript. The following points have to be addressed before meeting the journal standards:

1. Abstract : what  menas the following sentence :"The results show that MT has a wide range of 19 regulatory effects on plant growth indicators under salt stress, of which the regulatory effect on root 20 indexes is the strongest, and this regulatory effect is not specific-specific" does the word specific-specific menas species-specific?

2. Material and Methods section: Please explain duplicates removal in the screening of scientific papers. The procedure for validation of manuscripts is not enough clear in my point of vue. I suggest either rewriting this section more precisely or adding a supplementary material data that explians more widely and deeply the selection procedure and gives examples of studies for each criteria used, This is in my point of vue critical for the readers to trust the results and validity of the mataanalysis.

3. why  no workflow management or platform/software was used in the selection of manuscripts is this a choice or du to a certain limitation of the study.

4. In the extarction step of data have the results of extraction softwares checked by reviewers.

5. In section "3.3 The response of plant types to exogenous MT" what means the word plant types ? is it plant species? please be precise 

6. In discussion section please explain why "MT significantly (P < ≤0.05) promoted 256 the growth parameters of 12 plants except Avena sativa, Brassica napus, Helianthus annuus, 257 Hulless barley, Malus hupehensis and Stevia rebaudiana Bertoni." this is critical to abe addressed. than you claim that "4.1 MT has no species specificity in regulating plant growth under salt stress" Actually MT has species specificity.

7. why there is no report on effect of MT on plant microbiome reported in this study ? actually the manucript doi: 10.3389/fmicb.2019.02616 describes such an effect!

8. Conclusion section is very weak please discuss widely your findings what are the main findings and how this can be translated in practical recommandations. What are the current gaps of research in the field. What are the future directions and experiments that needs to be conducted.

I am really waiting to read an improved version of this manuscript that addresses all these drawbacks and that I could recommand for publication.

Best regards

Author Response

Reviewer 2

Dear authors,

Manuscript antioxidants-1855178 entiteled "Basic cognition of melatonin regulation of plant growth under salt stress: A meta-analysis" and authored by Feiyu Yan , Hongliang Zhao , Longmei Wu , Zhiwei Huang , Yuan Niu , Bo Qi , Linqing Zhang , Song Fan , Yanfeng Ding , Ganghua Li * , Guoliang Zhang targets a hot topic that is potentially very interesting for the journal readers, the whole scientific community and end users. While the work seems to be accurately conducted several issues needs attention of the authors that needs to provide clear answers that will determine the fate of the manuscript. The following points have to be addressed before meeting the journal standards:

Response: Thank you for your work in reviewing our manuscripts. We sincerely apologize for the mistake caused by our negligence. In the re-submitted manuscript, we carefully considered your proposal and corrected the errors you pointed out. We hope you can review our manuscript again and get your positive comments. Thank you again for your work.

  1. Abstract : what  menas the following sentence :"The results show that MT has a wide range of 19 regulatory effects on plant growth indicators under salt stress, of which the regulatory effect on root 20 indexes is the strongest, and this regulatory effect is not specific-specific" does the word specific-specific menas species-specific?

Response: We apologize for our inappropriate statement. Our study found that melatonin has a wide range of regulatory effects on growth related indicators such as dry weight, fresh weight, root length and leaf area of plants under salt stress. Therefore, we believe that there is no specificity in the regulation of melatonin on plant growth indicators. We grouped the growth indicators into shoot related indicators and root related indicators. The results of subgroup analysis showed that melatonin had the strongest regulatory effect on root related indicators. In the revised manuscript, we corrected the incorrect use of "specific-specific".

  1. Material and Methods section: Please explain duplicates removal in the screening of scientific papers. The procedure for validation of manuscripts is not enough clear in my point of vue. I suggest either rewriting this section more precisely or adding a supplementary material data that explians more widely and deeply the selection procedure and gives examples of studies for each criteria used, This is in my point of vue critical for the readers to trust the results and validity of the mataanalysis.

Response: Thank you very much for your suggestions. In the revised manuscript, we have perfected and enriched this part of the content to make the description more accurate. For some key parameters, we added references. In table A1, we give details of all the literature included in the study.

  1. why  no workflow management or platform/software was used in the selection of manuscripts is this a choice or du to a certain limitation of the study.

Response: Thank you for pointing this out. This is our statement that workflow management is not used, and this does not mean that there are limitations in this study.

  1. In the extarction step of data have the results of extraction softwares checked by reviewers.

Response: Thank you for your suggestion. We provided the extracted raw data for the reviewer to check. Please see the attachment.

  1. In section "3.3 The response of plant types to exogenous MT" what means the word plant types ? is it plant species? please be precise 

Response: We apologize for our inappropriate statement. We have corrected this error in the revised manuscript.

  1. In discussion section please explain why "MT significantly (P < ≤0.05) promoted 256 the growth parameters of 12 plants except Avena sativa, Brassica napus, Helianthus annuus, 257 Hulless barley, Malus hupehensis and Stevia rebaudiana Bertoni." this is critical to abe addressed. than you claim that "4.1 MT has no species specificity in regulating plant growth under salt stress" Actually MT has species specificity.

Response: Thank you for your suggestion. Melatonin regulation of plant salt tolerance has been widely reported, which covers a large number of plant species. In our study, through the subgroup analysis of plant species, the results show that melatonin has a significant regulatory effect on some species, and the family and genus relationships of these species are very far. We believe that this result shows that there is no species specificity in melatonin's regulation of plant growth under salt stress. However, as you pointed out, in the results of subgroup analysis, melatonin did not play a significant role in regulating the growth of some species under salt stress, which may be caused by the increase of confidence interval due to the small number of included studies. In addition, there is no significant difference between the effect values of most species, which also proves that melatonin has no species specificity for the regulation of salt stress on plant growth. We have retained this part in the revised manuscript and hope to get your permission. Thank you again for your work during the review of the manuscript.

  1. why there is no report on effect of MT on plant microbiome reported in this study ? actually the manucript doi: https://doi.org/10.3389/fmicb.2019.02616 describes such an effect!

Response: Thank you for your suggestion. In this study, we mainly focused on the regulation of plant growth by melatonin, so we did not study the contents related to plant microbiome. The functions of melatonin in algae, forge, wild animals and humans were reviewed in detail (https://doi.org/10.1016/j.envres.2021.111746). Thank you very much for providing this research. We have included it in the references, which helps deepen our introduction and discussion.

  1. Conclusion section is very weak please discuss widely your findings what are the main findings and how this can be translated in practical recommandations. What are the current gaps of research in the field. What are the future directions and experiments that needs to be conducted.

I am really waiting to read an improved version of this manuscript that addresses all these drawbacks and that I could recommand for publication.

Response: Thank you for your suggestions. In the revised manuscript, we revised the conclusion in detail to make it more in-depth and extensive, and the results obtained are more practical.

Round 2

Reviewer 2 Report

Dear authors,

I appreciate your work addressing all my comments I could now suggest your manuscript for publication.

Best regards